# A Multi-Channel Frequency Router Based on an Optimization Algorithm and Dispersion Engineering

**DOI:** 10.3390/nano13142133

**Published:** 2023-07-23

**Authors:** Hongyi Yuan, Nianen Zhang, Hongyu Zhang, Cuicui Lu

**Affiliations:** Key Laboratory of Advanced Optoelectronic Quantum Architecture and Measurements of Ministry of Education, Beijing Key Laboratory of Nanophotonics and Ultrafine Optoelectronic Systems, School of Physics, Beijing Institute of Technology, Beijing 100081, China

**Keywords:** multi-channel, frequency router, topological optimization, synthetic dimension

## Abstract

Integrated frequency routers, which can guide light with different frequencies to different output ports, are an important kind of nanophotonic device. However, frequency routers with both a compact size and multiple channels are difficult to realize, which limits the application of these frequency routers in nanophotonics. Here, a kind of bandgap optimization algorithm, which consists of the finite element method and topology optimization, is proposed to design a multi-channel frequency router. Channels supporting photonic edge states with different frequencies are built through the synthetic dimension of translational deformation. Due to the help of the developed optimization algorithms, the number of channels and output ports can be increased up to nine while maintaining ultracompact device size. The device operates within a working band of 0.585–0.665 c/*a*, corresponding to 1.504–1.709 μm when the lattice constant is set as 1 μm, covering the telecom wavelength of 1.55 μm. The average crosstalk is about −11.49 dB. The average extinction ratio is around 16.18 dB. Because the bus of the device can be regarded as a part of a topological rainbow, the results show that the structure is robust to fabrication errors. This method is general, which can be used for different materials and different frequency ranges. The all-dielectric planar configuration of our router is compact, robust, and easy to integrate, providing a new method for on-chip multi-channel broadband information processing.

## 1. Introduction

With the rapid development of big data, the internet of things, and 6G communication technology, the requirements of optical communication technology are increasing [1,2,3,4,5,6]. Frequency routers, which can guide light signals with different frequencies to different ports, can be widely used in photonic computing [7], photonic communication [8], and photonic interconnection [9]. The more channels a frequency router has, the bigger information capacity the device has. Therefore, multi-channel frequency routers are very important in practical applications. There have been schemes of designing multi-channel frequency routers proposed before. Micro rings can be used to design frequency routers with multiple channels, but the footprint is large [10]. Multi-channels can also be achieved using arrayed waveguide gratings, but the device is bulky and not suitable for on-chip integration [11].

In recent years, algorithms have been introduced in the design of nanophotonic structures, which provide an efficient method for realizing nanophotonic devices with good properties [12,13,14,15]. Devices optimized with algorithms usually have both compact footprints and good performances. There are various nanophotonic devices that have been designed by different algorithms, including wavelength routers [16,17,18], polarization routers [19,20], power splitters [21,22,23], metasurfaces [24,25,26], and other devices [27,28,29,30]. The applied algorithms can be divided into different categories, including heuristic algorithms such as genetic algorithms [16,20], gradient-based algorithms such as topology optimization [31], and artificial intelligence such as neural networks [24]. These algorithms deal with the design problem by directly searching the optimal results in the parameter space or predicting the parameters of the wanted target, which is more efficient than the traditional strategy of trial and error [14,15]. 

There are frequency routers that have been optimized with genetic algorithms [16] and inverse design [17,18]. These optimization algorithms take the transmissions of the output ports as optimization targets and the distributions of the material as design variables, which is convenient for exploring the parameter space and obtaining good performances within a given footprint. However, it is still difficult to design a multi-channel frequency router with optimization algorithms.

In this work, we propose a method to design a multi-channel frequency router based on a bandgap optimization algorithm and dispersion engineering. First, topology optimization and the finite element method are combined to design wide-bandgap two-dimensional photonic crystals. Second, lattice translations are introduced into the optimized photonic crystals along the x direction and y direction to achieve dispersion engineering. The bandgap ranges from 0.508 c/*a* to 0.665 c/*a*. Here, c is the light speed in vacuum and *a* is the lattice constant of the photonic crystal. A nine-port frequency router in silicon-based material is constructed, exhibiting a broadband-information-routing ability. The working band is selected as 0.585–0.665 c/*a*, which corresponds to 1.504–1.709 μm when the lattice constant is set as 1 μm, with a bandwidth of 205 nm, covering the telecom wavelength of 1.55 μm. With nine channels, the footprint of the device is compact. When the lattice constant is set as 1 μm, the footprint is 20 μm × 50 μm. It is shown that the multi-channel frequency router is robust to small fabrication errors. The design procedure of the multi-port frequency router can be used for various target frequencies and materials. This work provides a method for realizing light routing and paves the way for on-chip broadband information processing.

## 2. Methods

The optimization algorithm for wide bandgap photonic crystals is built by combining a topology optimization algorithm and the finite element method. The key target of the algorithm is to enlarge the bandgap between the dipolar modes and quadrupolar modes of the photonic crystals [32,33,34], so that the photonic router can be designed to operate at more channels with more edge states in a larger operation band. This eigenvalue optimization problem can be simplified to a scattering optimization problem. As an analogy of nanocavities, the design of photonic crystals can adopt a similar strategy by introducing frequency-average local density of state (LDOS) to quantify the stimulation of an eigen mode. The LDOS can be calculated by integrating the energy radiated by a dipolar source in the primitive cell. The eigenmode stimulated by a dipolar source is the function of the location of the dipolar source. For wide-bandgap photonic crystals, the target frequencies for the dipolar modes and quadrupolar modes should be optimized by maximizing the energy radiated from the corresponding dipolar modes simultaneously. 

The topology optimization method used here is the density-based method, which is a general structure optimization method and has been widely applied in different fields including mechanics, acoustics, and photonics [35]. Density-based topology optimization defines the continuously variable refractive index on each finite element and pushes the indexes to the ends to form binary structures [36,37]. Because of the great number of finite elements generated in the finite element method, the quantity of the variables in the design is tremendous, which provides a large performance space to look for good results. The finite elements in a model for a photonic unit cell can be up to thousands, depending on the mesh density applied. Simply calculating the gradient for one variable at one time of the simulation is time consuming. To address this point, an adjoint method is adopted to conduct a gradient analysis [38]. The idea of this adjoint method is to obtain the gradient information of a system by calculating one time of a forward model and one time of a backward model. Compared with brutal force, when calculating the gradients in all the optimization directions one by one, the adjoint method provides gradients with only two times of simulation, no matter how many design variables there are. 

The flow chart of the optimization algorithm is shown in Figure 1. Before the optimization starts, the target frequencies, maximum iteration number, and optimization tolerance are specified. In step ①, a randomly generated structure is given as the initial guess. In step ②, predefined models are simulated with the finite element method and the LDOSs are calculated. The finite element simulation is achieved with the commercial software COMSOL Multiphysics. In step ③, the gradient information is calculated based on the electromagnetic fields extracted from the models in the former step. In step ④, the method of moving the asymptotes is used to update the variables, which has a good performance in optimization problems with a large number of design variables. In step ⑤, a filtering procedure is added to avoid extremely small inner structures that are difficult for fabrications. The Gaussian filter is used here to smooth the value of each element. In step ⑥, a binarization step is added to push the structure to a 0-1-type material composition during the iteration. This step is achieved by introducing a penalty term to the materials with a refractive index between two ends. The penalty term should consider two materials without bias. A Heaviside step function is used to write the penalty term. In step ⑦, to keep the symmetry property of the lattice, an operation is added to force the lattice to be symmetric. In step ⑧, the convergence condition is tested. If the maximum iteration number is reached or the difference between the target expression values of two adjacent iterations is smaller than the tolerance, the optimization stops. After the optimization, a photonic crystal with a large bandgap around a target center frequency will be generated. The optimized photonic crystal is then used to build a multi-channel frequency router using translational deformation.

Below, we simply introduce several typical algorithms used for the design of nanophotonic routing devices and make a comparison among our method and these algorithms. There are various related algorithms that have been proposed in recent years [12,14,15]. The genetic algorithm (GA) is a commonly used method to design nanophotonic devices. A wavelength router [16] can be designed with the assistance of the GA. In these works, the GA is employed to tune the positions and sizes of the inner geometries in the design area, which leads to achieving the target functions on small footprints. However, fixed patterns can usually be seen in the designed structures, which may limit the exploration of the performance in the parameter space [16]. Topology optimization algorithms [31,35,39] and the inverse design approach [17,18,40,41,42,43] are methods that can generate freeform structures, which leads to a larger performance space to explore. A topological wavelength router [44] and spatial wavelength-polarization router [45] can be designed with the assistance of topology optimization. The inverse design approach integrated density variable method, level set method, and adjoint method together have been proven efficient in the design of wavelength routers [17,18,46].

The former introduced algorithms are successful in designing nanophotonic wavelength routers. However, a multi-channel frequency router with a channel number of nine has not been reported to be designed with these methods, which may limit the on-chip optical information-processing capacity. Our design method designs a multi-channel frequency router by converting the problem from optimizing multiple channels directly to optimizing the band gap of the photonic crystal. With the optimized photonic crystal, a translational synthetic dimension is used to build a frequency channel to support photonic states with different frequencies. The role of our algorithm is to make the bandgap of the photonic crystal large enough to support multiple channels.

## 3. Results

The sketch map of the designed device is shown in Figure 2a. Light incident from the downward silicon waveguide will be guided to one of the nine output ports according to its frequency. Two kinds of interfaces are marked by blue and yellow solid lines. $A_{0}–A_{5}$ are the regions where the light is slowed, while $O_{1}$–$O_{9}$ are the output channels where the signals are output. The details of the designed device are shown in Figure 2b. The refractive indexes of the silicon and air are set as 3.45 and 1, respectively. The interfaces are formed by moving the photonic crystals on the side of the arrows with translational deformation along the corresponding arrows. A blue arrow marks the horizontal lattice translation, η. A yellow arrow marks the vertical lattice translation, ξ. The function of this frequency router is based on dispersion engineering at two photonic crystal interfaces. The dispersion curves of the edge states within the bandgap can be easily controlled by the lattice translation.

When η is increased gradually by a lattice period in the y direction, a Bloch wave vector, *k*, and η can form a closed torus in a synthetic space, and the eigenstates on the surface of the torus are topologically nontrivial [47,48]. A topological rainbow is constructed on the left side of Figure 2c. The parameter η changes from 0.86a to 0.88a with a step of 0.03a, which covers a lattice period. Five rows of holes are allocated to each value of η, except 0.86a, which occupies the first ten rows. Because the length of the rainbow is large, the structure details only at the beginning and the end are presented, while the structures in the middle are omitted here. The distribution of the electric field norm is shown on the right side of Figure 2c. The bus of the frequency router can be regarded as a part of the rainbow, which gives the designed device robustness. The robustness makes the device tolerant to small fabrication errors in the structure, which will be shown in the discussion. When the translational parameter ξ is 0.5a, topological edge states can be generated on the horizontal interfaces [49].

The photonic crystal used to build the frequency router is generated by the optimization algorithm, whose energy band diagram and structure are shown in Figure 2d. The bandgap lies between 0.508 c/*a* and 0.665 c/*a*, with a bandwidth of 0.157 c/*a*. Figure 2e shows the energy band diagram of a honeycomb lattice with the same materials, which has a narrower band gap than the optimized structure. The gap is of 0.021 c/*a*, ranging from 0.571 c/*a* to 0.592 c/*a*.

In this structure, the parameter η changes gradually along the y direction from region $A_{0}$ to region $A_{5}$, and the frequency range of the edge state allowed to propagate through these regions gradually decreases. To achieve a balance between the port number and device size, only a part of the energy band gap is used to build the frequency channels. From region $A_{0}$ to $A_{5}$, η takes 0.86a, 0.82a, 0.80a, 0.78a, 0.74a, and 0.72a. The dispersion curve of the edge state is shown in Figure 3a, along with the corresponding supercell below. Decreasing the lattice translation from 0.86a, the corresponding frequency of the eigenstates decreases. Region $A_{0}$ supports all the frequencies within the bandgap, while $A_{1}$ supports only the light with a frequency lower than 0.65 c/*a*. The supported frequencies of the different regions decrease along the direction of the y axis. If the incident light comes to a region that does not support its frequency, it is slowed and reflected. The maximum propagating distances of different frequencies are different. 

Another vertical lattice translation, ξ, is introduced to form nine channels, which are used to guide the light with different frequencies to the output ports. In Figure 3b, the edge states of the second supercell have a narrow frequency range. To match the operation frequencies of these two interfaces, the vertical lattice translation decreases from 0.44a to 0.36a, where the frequency of the second type of edge state decreases from 0.665 c/*a* to 0.585 c/*a*. For each output port, this interface works as a band-pass filter, supporting light with only the target frequency to propagate through it.

The frequency router is achieved by combining the above two interfaces. The transmission spectrum of this device is shown in Figure 4a. The transmission of each port is normalized by the transmission value of O4, the highest of the nine ports. It is easy to notice that most peaks in the transmission spectrum have a narrow full width at half maximum (FWHM), which is more remarkable at a low frequency range. There exists at least one peak transmission frequency for each output port, marked by a Roman number (I)–(IX). The frequencies of the output signals can be selected from the peaks at will by adding the external frequency filters. The energy density distributions of the electric field at all the nine peaks (I)–(IX) are shown in Figure 4b. Incident light travels along the y direction as far as possible, and then turns left or right to the output port. It is clear that the traveling modes in different areas are different from each other, demonstrating the mode conversions. The simulation is achieved with commercial finite element simulation software, COMSOL Multiphysics. By setting the lattice constant *a* as different values, a multi-channel frequency router in different frequency bands can be designed.

The narrow FWHM and multiple modes can be explained by the mode mismatch at the junction of two interfaces. Taking the junction between $A_{0}$ and $A_{1}$ as an example, light with a frequency lower than 0.65 c/*a* is not supported by two yellow interfaces of O8 and O9, so it keeps traveling along the y direction. However, blue interfaces in $A_{0}$ and $A_{1}$ hold different modes for light at the same frequency because of their different translation deformations, so there is a mode conversion when light crosses this junction along the y direction. This conversion efficiency is dependent on the frequency, which acts as a filter, forbidding some frequency ranges from the target range. The more junctions the light passes through, the more frequency bands are prohibited in the spectrum. For light with lower frequencies, the transmission spectrum remains with only one or two peak signals due to the filtering effects from all the junctions it passes through. This filtering effect helps to reduce the FWHM and induce multiple modes in each port, which can also be utilized to engineer the peaks in the transmission spectrum by optimizing the junction area.

The average crosstalk [50] of the channels is about −11.49 dB and the average extinction ratio [51] is around 16.18 dB. The loss of the device mostly comes from insertion loss, because the photonic crystals reflect and scatter a part of the energy when the photonic edge states are excited along the boundary. The average insertion loss [52] of the device is about 0.97 dB, which is bigger than what was reported by Piggot et al. [17] and Logan Su et al. [18]. The insertion loss can be compensated for in practical applications by adding an amplifier behind [53,54]. The channel number in this work is 9, more than these two works, which may provide a bigger information-processing capacity. Recently, a sub-wavelength grating (SWG) method has also been used to design routing devices [55], which is based on a thermal-optic interferometric configuration and can be used in the telecommunications field. Our device is based on optimized photonic crystals and process light with a frequency within the bandgap. To further improve the performance of the frequency router, it is possible to optimize the structure near the input ports and output ports of the router to reduce the loss when light propagates [56].

## 4. Discussion

To test the robustness of this frequency router, disorders were introduced into the structure to mimic the defects caused by lithography imperfections [57,58,59]. Without a loss of generality, we chose the path for peak frequency (V), and studied the influence induced by these disorders. As shown in Figure 5a, the lattices were shifted, taken away, or deformed in four disorders along the path where the light with a peak frequency (V) travelled through. The distributions of the electric field amplitude before and after introducing the disorders are shown in Figure 5b,c, respectively. The propagation of the peak frequency (V) was basically maintained. The robustness of the device can be explained from the angle of the topological rainbow, because the main bus of the frequency router was a part of the topological rainbow, as shown in Figure 2c. Comparing Figure 5c with Figure 5b, it can be seen in the disturbed structure that there were some scatterings along the channel, especially the corner where the light turned into the output port. This can be explained by the mode mismatch in the junction with different translational deformations. If the disorder strength was too large, the device would not be able to work normally. The shifts of the unit cells were smaller than half of a lattice constant. The translation parameter was kept undisturbed in the test of robustness, which promised the basic function of the device. The function of our device relied on the designed frequency ranges supported by the structures with gradually changing translational parameters. Two structures with adjacent translational parameters had a common part in their supported frequency ranges, which allowed the light within the common frequency range pass through to the next channels, while the light outside of the common frequency range was stopped here and turned into the corresponding ports. If the added disorders were exactly on the bus channel and changed the translational parameters, the supported frequency range near the perturbed structures could be shifted. The light with target frequencies of the output ports could fly to the next ports or be stopped near the former ports because of the unwanted frequency range shifts. As consequence, the transmissions of the output ports near the disorders could be decreased and the output ports in the next channels could be even closed. 

The minimum of the differences in the translational parameters in our device was 10 nm when the lattice constant was chosen as 1 μm, which could cause very small details of about several nanometers on the channels. In real fabrication, these details can be etched with electronic beam lithography. The occasional small errors brought about by the fabrication away from the channels and junctions would have little harm on the performance of the device. Errors exactly on the channels should be decreased as much as possible. As discussed in the part of robustness test, the frequency shifts brought about by the defects in the channels would cause fatal deterioration to the performance of the device. Under this kind of circumstance, the signals from the ports near the defects could be decreased and some channels could even be shut off. In practical experiments, a scanning optical near field microscope can be used to measure this intensity distribution in a device [48]. To measure the transmissions of the nine ports, coupling gratings and waveguides should be added. The coupling gratings were used to input and receive signals with fibers. Noises from the environment are always inevitable, which may bring about bad effects on detecting the signals of our frequency router. Considering the coupling efficiency between the fibers and gratings, the signals would be small when detecting directly. Thus, it is necessary to fabricate a sample with a blank structure under the same fabrication conditions, in order to measure the transmissions of both samples and compare the transmissions of the devices to the transmissions of the blank sample. This way, the noise from the environment can be suppressed when measuring the device. 

## 5. Conclusions

In conclusion, we proposed a new method of designing multi-channel frequency routers based on optimization algorithms. The photonic edge states were achieved based on a synthetic dimension through translational deformation. The optimization algorithm consisted of topology optimization and the finite element method. By using the optimization algorithm, a photonic crystal with a wide bandgap was designed and the output port number of the frequency router was increased up to nine. The bandgap ranged from 0.508 c/*a* to 0.665 c/*a*. The designed device worked in the band of 0.585–0.665 c/*a*, corresponding to 1.504–1.709 μm when the lattice constant was set as 1 μm, with a bandwidth of 205 nm, covering the telecom wavelength of 1.55 μm. The footprint was 20 μm × 50 μm when the lattice constant was set as 1 μm. The frequency router was also robust to fabrication errors. The device was compact and the method could be applied to different materials and different frequency ranges. This work provides a new method for on-chip routing and broadband information processing.

## Figures and Tables

**Figure 1 nanomaterials-13-02133-f001:**
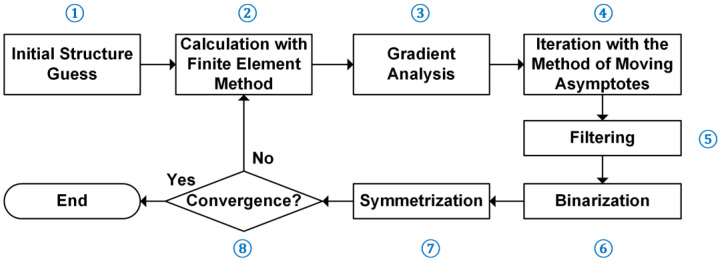
The flow chart of the optimization algorithm. Finite element method is employed to simulate the photonic crystal. The part of topology optimization includes steps of iteration, filtering, and binarization. The step of symmetrization is for aim of keeping the symmetry property of the photonic crystal.

**Figure 2 nanomaterials-13-02133-f002:**
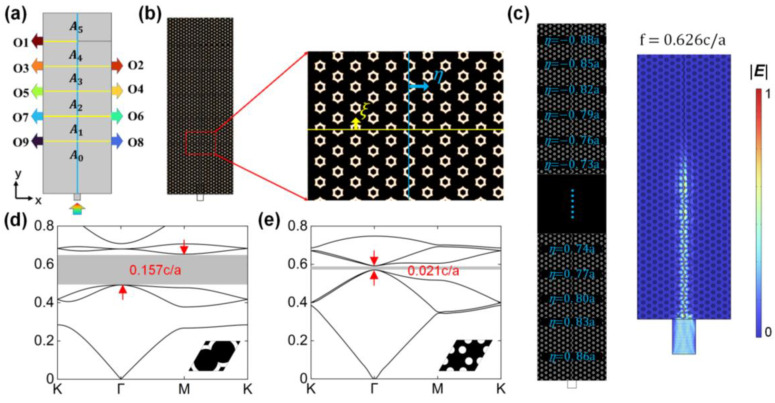
(**a**) Schematic diagram of the multi-channel frequency router. Two photonic crystal interfaces are marked by blue and yellow solid lines. Signals with different frequencies input from the lower port and output from the corresponding ports marked by arrows with different colors. (**b**) General view and enlarged view of the frequency router. Silicon is in black and air is in white. (**c**) Schematic diagram of the topological rainbow (**left panel**) and distribution of the electrical field norm at frequency of 0.626 c/*a* (**right panel**). (**d**) The energy band diagram of the optimized structure. (**e**) The energy band diagram of honeycomb lattice without optimization.

**Figure 3 nanomaterials-13-02133-f003:**
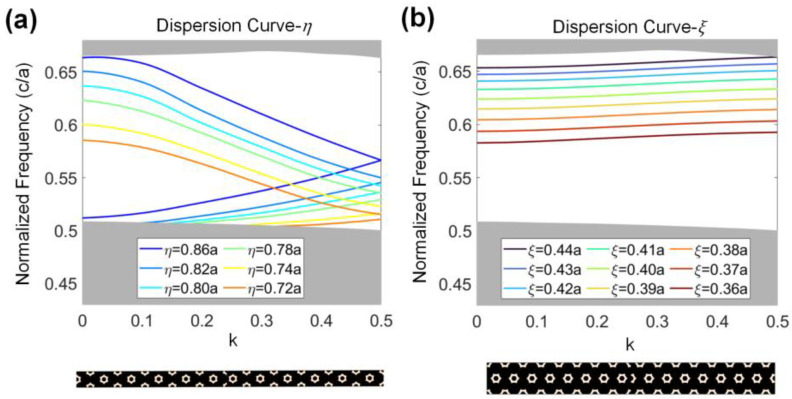
(**a**) Dispersion relationships under the horizontal lattice translation, η. Supercell of interface with η is shown below. (**b**) Dispersion relationships under the vertical lattice translation, ξ. Supercell of the interface with ξ is shown below. The supercells in both figures are formed by moving the lattice on the right side with given translational parameters. The parameter k is the normalized wavevector in the first Brillouin zone of the unit cell.

**Figure 4 nanomaterials-13-02133-f004:**
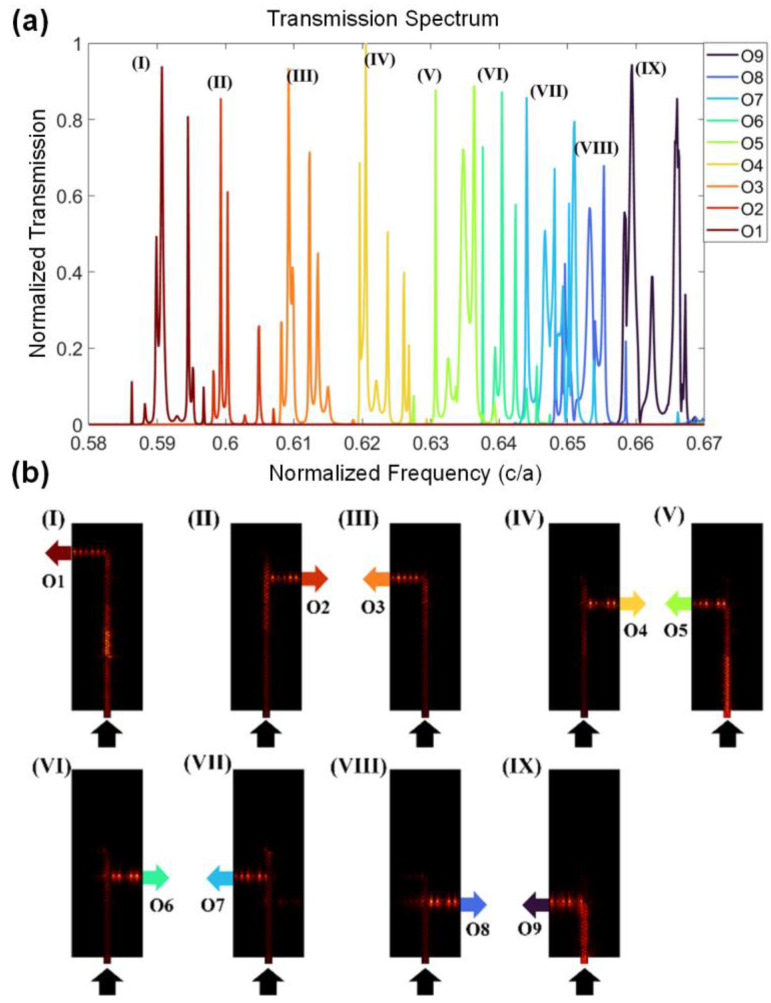
(**a**) Transmission spectrum of the designed frequency router. A peak for each output port is marked by Roman number I–IX, respectively. The peak frequencies are 0.591, 0.599, 0.609, 0.620, 0.631, 0.640, 0.644, 0.655, and 0.660 c/*a*. (**b**) The energy distribution of electric field at these nine peak transmission frequencies.

**Figure 5 nanomaterials-13-02133-f005:**
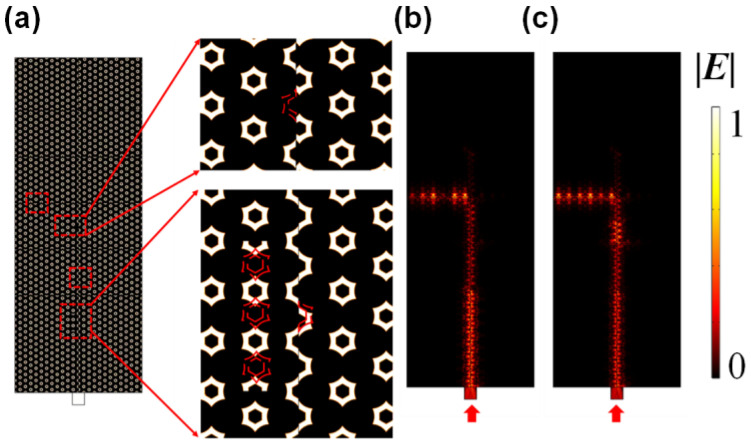
(**a**) Schematic diagram of the proposed structure with disorders. The hexagonal-ring shaped scatters in the red dashed box are shifted, moved away, or deformed. (**b**,**c**) The distributions of electric field amplitude before and after introducing disorders.

## Data Availability

The data presented in this study are available on proper request from the corresponding author.

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
