# Peer review of "A Multi-Channel Frequency Router Based on an Optimization Algorithm and Dispersion Engineering"

_nanomaterials, 2023, doi:10.3390/nano13142133_

Round 1

Reviewer 1 Report

The manuscript entitled "Multi-channel frequency router based on optimization algorithm" describes the finite element method and topology optimization to design a multi-channel frequency router. However, a few comments and suggestions must be addressed before this reviewer recommends the publication of this work in the Journal.

Comments

1.       Abstract section should improve with important results, like parameters and frequency ranges.

2.       There is no references in the discussion section. The author should provide the suitable references in discussion section.

3.       The author should revise the outstanding points (bandgap and bandwidth values) and highlights of this work in the conclusion.

4.       Typographical errors and superfluous spaces throughout the manuscript should be corrected.

Author Response

Dear reviewer, thank you for your valuable comments and suggestions, please see the attachment for the response.

Reviewer 2 Report

In the manuscript, there are shown results of optimization and simulation of generalized frequency router based on photonic crystal effect. The content and the results of the manuscript are very interesting, but the results are not presented clearly enough. At least for me there is a lot of information missing to fully understand the results presented.

For example, I am missing the information about the c and a parameters in the whole content. I understand that a parameter is somehow related to the period of the photonic structure, and c parameter is related to the frequency of the transmitted wave, but the exact relation is hidden. Moreover, as much as I tried, I am not able to decode the k parameter in Fig. 3. What is it related to and how?

In final parts of the manuscript are shown simulations of the whole device with resulting "spectral" graphs for outputs and energy distributions for each outputs. I suppose that it has been simulated using some of the available software tools, but I could not find the information in the content. Moreover, the results are shown very general for normalized frequencies but I can imagine that for very low frequencies in THz range would the output spectra look different in comparison to UV frequencies. Was the simulation done just for normalized frequencies or did you simulated the spectra of outputs using exact a and c parameters?

Due to the mentioned lack of information I cannot suggest to accept the manuscript in the current state, but I will suggest to accept it after minor revision when the content will be more clear to the reader.

Author Response

(The authors gave the same response as above.)

Reviewer 3 Report

The proposed study proposes a kind of bandgap optimization algorithm consisting of finite element method and topology optimization for multi-channel frequency router design. The algorithm is optimized, claiming to increase the number of channels and output ports up to 9 with the size of a tiny device. It also claims to be resistant to fabrication errors and advantageous for integration. However, in order to publish the paper, the following problems must be corrected and supplemented.

1. The title is too broad compared to the proposed study, so it needs to be modified to suit the proposed study.

2. In the second section, it is necessary to introduce related research such as existing nanophotonic structures and existing algorithms. It should also explain what advantages and disadvantages they have.

3. Next, it should clearly show how the proposed algorithm overcomes their limitations instead of various existing algorithms.

4. To check how advanced the proposed study is and how well it has good performance, The authors must add parts that can be compared with existing studies in any way, whether qualitative or quantitative.

Some typing mistakes must be corrected through reading

Author Response

(The authors gave the same response as above.)

Reviewer 4 Report

The reviewed manuscript reports on study of a multi-channel optical router using a two-dimensional photonic crystal. This material elicits the following comments:

1. A router is a device for redirection of data packets according to certain rules between the input and output ports. The main parameters of a router would be the switching speed, losses inside the device, and so on. It is necessary to specify these parameters of the proposed router and to compare them with other solutions. Also, it is necessary to point out the advantage of the proposed optimisation algorithm.

2. It should be explained how exactly redirection switching is performed (electrically, mechanically?). What is the advantage of the proposed switching method?

3. The weak point of on-chip routing is the technology of guiding the radiation into and out of microscopic ports. It is necessary to comment improvement of the router itself as separate from the inter-connect technology.

If the provided comments are taken into consideration when working on a new revision of the manuscript, it may be published in Nanomaterials

Author Response

(The authors gave the same response as above.)

Reviewer 5 Report

Report on:

 Multi-channel frequency router based on optimization algorithm

by  Cuicui Lu et at 

The paper presents a new method to implement a frequency router using a photonic crystal. This is a timely research idea that might potentially add a visible contribution to this study domain. Unfortunately, the way the research is presented in the paper doesn’t provide any means of evaluating the real advancements made by the authors in their project. I would encourage authors to rewrite the paper and concentrate on description of their device and its performance, highlighting important achievements.

Here are my main critical comments:

·      Section 2 provides very basic introduction into classical electrodynamics and differential equation solution that can be found elsewhere. Please, describe the methodology of research and give a reference to textbooks, if needed, instead.  

·      Section 3 contains only simulation results that are achieved under ideal conditions. A very short and superficial discussion of the router robustness in Section 4 is neither sufficient nor convincing. In general, one would need to demonstrate the method in real implementation to be able to claim “new method” (section 5 lines 239-242 ). If it is not feasible to manufacture a prototype, discuss the needs and limitations of the current technology to make it true. Provide a detailed study of the performance deterioration under realistic conditions.

Minor comments:

·      Figures captions are not informative

·      Figures sizes are too small and quality is too poor for understanding the research results

I recommend to send the article back to authors for amendments and improvements. In its current form it should not be published.

Author Response

(The authors gave the same response as above.)

Reviewer 6 Report

The Authors propose a multi-channel frequency router based in optimization algorithm based on 2D-PhC and the exploitment of topological states. The reported results have been achieved by using FEM simulations. Here, my comments to the manuscript:

-          The proposed paper should be classified as Communication.

-          The Authors reports that their aim regards the engineering of the dispersion. However, besides topological photonics, subwavelength grating (SWG) fulfils the requirement. Moreover, several SWG-based configurations have been proposed also useful for routing (see, i.e., Design of a large bandwidth 2× 2 interferometric switching cell based on a sub-wavelength grating. Journal of Optics23(8), 085801, 2021).

-          The Authors use a translation parameter of 0.50a to excite topological states. Please correlate it to the fabrication suitability.

-          Please improve the quality of Fig. 3.

-          In order help reader in the performance rating, please report insertion loss, crosstalk and extinction ratio for the whole router.

Author Response

(The authors gave the same response as above.)

Round 2

Reviewer 3 Report

The revised manuscript appears to have been well-modified according to the reviewer's request, so it is recommended to publish it as a paper.

Author Response

Thanks for your comments.

Reviewer 4 Report

In response to my observations, important information was added to the manuscript that made it more interesting and comprehensible. My comments have been fully addressed by the Authors in  the revised manuscript, which may be now published.

Author Response

Thanks for your comments.

Reviewer 5 Report

Report on:

 Multi-channel frequency router based on optimization algorithm

by  Cuicui Lu et at

version 2

I appreciate the work the authors did to implement my comments and suggestions on the first draft of this paper. Unfortunately, the team decided to implement only small textual changes without rewriting the presentation of the research.  My concerns remain the same: the way the research is presented in the paper doesn’t allow to judge on scientific soundness of the proposed methodology. At this point of review, I recommend to reject the paper.

Here are reiterate my main critical comments from the first round with some extra thoughts on the second version:

·      Section 2 provides very basic introduction into classical electrodynamics and differential equation solution that can be found elsewhere. This part should be either completely removed or radically shortened. Instead, a detailed description the research methodology should be given.

·      Section 3 contains only simulation results that are achieved under ideal conditions. A very short and superficial discussion of the router robustness in Section 4 is neither sufficient nor convincing. In general, one would need to demonstrate the method in real implementation to be able to claim “new method”. If it is not feasible to manufacture a prototype, discuss the needs and limitations of the current technology to make it. Provide a detailed study of the performance deterioration under realistic conditions.

Author Response

Dear reviewer, thank you very much for the comments and valuable suggestions. Please see the attachment about the response.

Reviewer 6 Report

The Authors have modified the manuscript according to the Reviewer suggestions.

Author Response

Thanks for your comments